# Germline burden of rare damaging variants negatively affects human healthspan and lifespan

**Anastasia V Shindyapina[1†], Aleksandr A Zenin[2,3†], Andrei E Tarkhov[2,4], Didac Santesmasses[1], Peter O Fedichev[2,5‡]\*, Vadim N Gladyshev[1‡]\***

[1]Brigham and Women's Hospital, Harvard Medical School, Boston, United States; [2]Gero LLC, Moscow, Russian Federation; [3]The Faculty of Bioengineering and Bioinformatics, Lomonosov Moscow State University, Moscow, Russian Federation; [4]Skolkovo Institute of Science and Technology, Skolkovo Innovation Center, Moscow, Russian Federation; [5]Moscow Institute of Physics and Technology, Moscow, Russian Federation

**Abstract** Heritability of human lifespan is 23–33% as evident from twin studies. Genome-wide association studies explored this question by linking particular alleles to lifespan traits. However, genetic variants identified so far can explain only a small fraction of lifespan heritability in humans. Here, we report that the burden of rarest protein-truncating variants (PTVs) in two large cohorts is negatively associated with human healthspan and lifespan, accounting for 0.4 and 1.3 years of their variability, respectively. In addition, longer-living individuals possess both fewer rarest PTVs and less damaging PTVs. We further estimated that somatic accumulation of PTVs accounts for only a small fraction of mortality and morbidity acceleration and hence is unlikely to be causal in aging. We conclude that rare damaging mutations, both inherited and accumulated throughout life, contribute to the aging process, and that burden of ultra-rare variants in combination with common alleles better explain apparent heritability of human lifespan.

**\*For correspondence:**
peter.fedichev@gero.ai (POF);
vgladyshev@rics.bwh.harvard.edu
(VNG)

[†]These authors contributed
equally to this work
[‡]These authors also contributed
equally to this work

**Competing interest:** See
page 15

**Reviewing editor:** Sara Hagg,
Karolinska Institutet, Sweden

## Introduction

Genome-wide association studies (GWAS) of human lifespan, including studies examining extreme lifespan, parental survival, and healthspan, produced a number of gene variants potentially associated with human aging. For example, GWAS on centenarians consistently demonstrate loci near *APOE* gene to be associated with extreme longevity, and loci near *FOXO3A*, *HLA-DQA1* and *SH2B3* genes to have population-specific associations (*Melzer et al., 2020*). However, even in developed countries, centenarians represent less than 0.1% of the population, and the genetic determinants responsible for the survival of general population remain poorly understood. Release of massive genotype and phenotype data by UK Biobank (UKB) (*Bycroft et al., 2018*) allowed to investigate the relationship between genetics and several longevity proxies, such as parental lifespan (*Pilling et al., 2017*) and healthspan, within the general population (*Zenin et al., 2019*). They confirmed most of the variants from centenarian studies and identified additional variants. However, the combined contribution of common variants could explain only a small fraction of the lifespan variation as most of the individuals lack any of the alleles previously associated with lifespan. We hypothesized that some of the remaining heritability could be explained by the combined burden of rare damaging gene variants as those are present in every genome. Until very recently, only common variants could be probed in genetic studies due to sample size limitations. However, large datasets such as gnomAD and UKB now allow assessing the effects of variants with minor allele frequency (MAF) lower than 0.1% (*Lek et al., 2016*).

**eLife digest** Most living things undergo biological changes as they get older, a process that we generally refer to as aging. Despite being a widespread phenomenon, scientists do not fully understand why we age, though it appears that a combination of genetics and lifestyle factors, such as diet, play a role in influencing lifespan. Aging increases the risk of developing a wide range of diseases, including cancer, Alzheimer's disease and diabetes. As such, finding ways to slow the aging process would help to postpone the onset of illness and potentially improve health in old age.

Genes are thought to be responsible for between one quarter and one third of the variation in human lifespans. The relationship between genes, aging and lifespan is complex and not well understood. One set of rare genetic changes that have been shown to have significant effects on diseases are called protein truncation variants (PTVs). PTVs cause damage by altering the production of certain proteins. There are many possible PTVs and people can be born with them or they can develop them in some cells later in life. The full influence of PTVs on aging is not known.

Shindyapina, Zenin et al. have now studied observational data collected from two groups of over 40,000 people in the UK. Both groups recorded over 1,000 deaths, and the study examined the influence of PTVs on natural lifespan. The results show that each person is born with an average of six PTVs, which can vary in the impact that they have on aging. Having more, or more severe, PTVs could reduce life expectancy on average by 1.3 years. PTVs affect both total lifespan and healthy lifespan, the period of time lived prior to developing the first age-related disease.

While PTVs that people are born with have a significant effect on aging, this study also showed that PTVs that are acquired due to spontaneous mutations through a person's life have much less of an impact. This is a key insight into the relationship between genes and aging. These discoveries could help in using genetics to anticipate future health, it also helps to identify some of the biological systems that have a role in aging. This could lead to new ways to delay the aging process and its effects on health.

These ultra-rare variants, most notably protein-truncating variants (PTVs), are known to be enriched for damaging alleles. They tend to have larger effect sizes and dramatically change gene expression and function. An inverse relationship between variant's minor allele frequency (MAF) and effect size was recently demonstrated for type II diabetes, an archetypal age-related disease (*Mahajan et al., 2018*). Multi-tissue gene expression outliers were enriched with rare variants in the GTEx dataset (*Li et al., 2017*). Notably, PTVs represent a significant fraction of those variants. Most of the underexpressed outliers harbor rare PTVs, which are more likely to trigger nonsense-mediated mRNA decay (NMD) than common variants (*Rivas et al., 2015*). Additionally, ExAC consortium demonstrated that the nonsense variants with a high Combined Annotation Dependent Depletion score (*Rentzsch et al., 2019*), a widely used predictor of deleteriousness of single-nucleotide variants, were enriched in singletons (*Lek et al., 2016*). Although missense and non-coding variants may also be damaging, PTVs are substantially enriched for deleterious alleles. They also alter gene expression more dramatically than missense and untranslated region (UTR) variants (*GTEx Consortium et al., 2017*).

Ultra-rare PTVs are usually eliminated by purifying selection, but the small effective population size of humans means that they are present in all human genomes. Increased rare PTV burden was associated with complex diseases, such as schizophrenia, epilepsy and autism (*Leu et al., 2015*; *Singh et al., 2017*; *Ji et al., 2016*), whereas individual genetic variants exhibited small effects. The burden of rare PTVs in genes intolerant to such variants was tested for association across ExAC traits, wherein these variants were defined as PI-PTVs - protein-truncating variants in proteins intolerant to protein-truncating variants (defined as having pLI score of 0.9 or above). This analysis revealed a negative association with years of schooling (academic attainment) and a positive association with intellectual disability, autism, schizophrenia and bipolar disorder (*Ganna et al., 2018*). Notably, the age at enrollment was also negatively correlated with the burden of PI-PTVs, suggesting a possible association with lifespan. Rare variants emerged as a novel genetic component with profound effect on complex traits and fitness. In this study, we focused specifically on the association of germline

PTV burden with lifespan and disease-free survival, and estimated the effect of somatic PTV accumulation on mortality and morbidity acceleration.

## Results

### Study design and data sources

We characterized the effects of inherited mutations burden on human traits associated with lifespan. For the UK Brain Bank Network (UKBBN), we ran a survival analysis against the age at death (*Keogh et al., 2017*). For UKB subjects, we tested the effects of mutations on lifespan and healthspan. For these analyses, we define lifespan as survival within a follow-up period of 11 years, and healthspan as the disease-free period before one of the following conditions is diagnosed for the first time (*Table 1*): cancer, diabetes, myocardial infarction, congestive heart failure, chronic obstructive pulmonary disease, stroke, dementia, and death (*Zenin et al., 2019*). In addition, following the approach of *Joshi et al., 2016*, we studied the effect of mutation burden on parental survival (separately for the age at death for mothers and fathers), a useful lifespan proxy in genetic studies.

We selected a cohort of 40,368 individuals from UKB with sequenced exomes who self-reported 'White British' and were of close genetic ancestry based on a principal component analysis of their genotypes (*Bycroft et al., 2018*). Of those, 21,742 (54%) were males with mean age of 58.1 years (SD = 7.9, age range $40.2 - 70.6$) and 18,626 (46%) were females with mean age of 57 years (SD = 7.8, age range $40.1 - 70.4$) at the time of assessment. In the UKB cohort, 1,122 subjects died during the follow-up period of 11 years ($2005 - 2016$), mostly of cancer (*Table 2*). The UKBBN cohort included 1,105 deceased subjects of European origin after we excluded cases of suicides, accidents, and cases of death with no abnormalities detected. Of those, 489 (44%) were females with mean age of 71.2 years (SD = 18, age range $16 - 103$ years) and 616 (56%) were males with mean age of 67.7 years (SD = 17, age range $17 - 105$ years). The cause of death was reported for 359 out of 1,105 individuals used for downstream analysis. Most participants in this study were diagnosed with neurodegenerative diseases, for example Alzheimer's, Parkinson's, and Pick's diseases (*Keogh et al., 2017*).

We used here the set of variants identified through whole-exome sequencing (WES) as part of the UKB and UKBBN projects (*Keogh et al., 2017*). As in *Ganna et al. (2018)*, we limited our analysis to PTVs, defined as splice donors/acceptors, stop codon gains, and frameshifts, observed in canonical transcripts. To address the relationship between the PTVs allele frequency and their effects on lifespan, we binned the PTVs according to their minor allele frequency: (1) $MAF<10^{-4}$; (2) $10^{-4}<MAF<10^{-3}$; (3) $10^{-3}<MAF<0.01$; (4) $0.01<MAF<0.2$. For each allele frequency bin, we computed the PTV burden as the total number of PTVs per individual's exome (*Figure 1—figure supplement 1* for PTV burden distribution in the MAF bins).

**Table 1.** Incidence of first disease (end of healthspan) statistics in UK Biobank subjects.
MI - myocardial infarction, COPD - chronic obstructive pulmonary disease, CHF - congestive heart failure.

|  | Number of events |
| --- | --- |
| Cancer | 6239 |
| Diabetes | 2009 |
| MI | 1862 |
| COPD | 619 |
| Stroke | 527 |
| Dementia | 211 |
| Death | 126 |
| CHF | 114 |

**Table 2.** Cause of death reported for 1,122 and 359 subjects in UKB and UKBBN cohorts, respectively.
UKB - UK Biobank, UKBBN - UK Brain Bank Network.

|  | UKB | Ukbbn |
|---|---|---|
| Neoplasm | 638(56.9%) | 20(5.6%) |
| Circulatory system | 208(18.5%) | 90(25.1%) |
| Respiratory system | 82(7.3%) | 171(47.6%) |
| Digestive | 47(4.2%) | 7(1.9%) |
| Nervous system | 43(3.8%) | 51(14.2%) |
| External | 35(3.1%) | — |
| Other (infections, congenital, endocrine, mental) | 69(6.1%) | 19(5.3%) |

## Survival analysis

We examined the association of PTV burden against lifespan traits (i.e. survival in UKB and UKBBN, the chronic disease free survival (healthspan), mother's and father's age at death in UKB) using variations of Cox proportional hazards (PH) models. We used sex, assessment center, and genetic principal components as covariates to account for the effects related to the population heterogeneity.

As we shall see below in the 'Somatic mutations and mortality acceleration' section, predicted effects of PTVs accumulation with age provides negligible contribution to mortality acceleration. Therefore, time-varying effects can be neglected and the age-independent PTV load contribution to the age-dependent mortality in UKB can be found by means of the standard Cox proportional hazards model (hereinafter referred to as the 'mortality risk' or survival model) using the follow-up survival information and the age of the first assessment as a covariate.

The survival model involving the follow-up time and the explicit age as the regression parameter is a maximum likelihood estimator of probability of short-term survival for the individuals healthy enough to survive till the age of the first assessment. In this form, the survival model does not depend on the life history and hence should be robust with regard to enrollment bias effects.

In *Zenin et al. (2019)*, we observed that the incidence of major chronic diseases (such as Congestive Heart Failure (CHF), Myocardial Infarction (MI), Chronic Obstructive Pulmonary Disease (COPD), stroke, dementia, diabetes, cancer, and death) in UKB increases exponentially with age in accordance to Gompertz law. Therefore, the end of healthspan can also be naturally modeled with the help of PH models including risk estimates exponential in age variables. As much as 28% of UKB participants were diagnosed with at least one of the selected diseases by the time of the first assessment. Therefore, the chronic disease-free survival, also known as healthspan, cannot be studied with the help of the standard Cox PH model.

Instead, in *Zenin et al. (2019)* we noted the very limited number of death events during the follow-up time in UKB and hence assumed that the incidence of diseases do not considerably affect enrollment. Accordingly, we suggested and employed here the maximum likelihood formulation of PH model (hereinafter referred to as the 'morbidity risk', simply 'morbidity', or healthspan model) involving the age at the first incidence chronic disease or the end of the follow-up time.

The mortality and morbidity risk models returned Cox regression parameters that were consistent with well-established mortality and morbidity acceleration patterns. For example, the survival (the remaining lifespan) model in the UKB produced the regression coefficient $\Gamma = 0.087$ (95 % CI 0.078–0.097) per year for the age of first assessment, very close to the mortality and morbidity acceleration rate of approximately 0.09 per year in UKB cohort (*Zenin et al., 2019*). The characteristic time scale is $t_{1/2} = \ln(2)/\Gamma = 7.5$ years and hence is nothing else but the mortality rate doubling time from Gompertz mortality law.

The Cox regression coefficients for males were 0.47 (95% CI 0.35–0.59) in UKB and 0.26 (95% CI 0.13–0.38) in UKBBN. Under constant mortality acceleration, this would correspond to approximately $3 - 5$ years of difference in life expectancy. Women in the UK (the population relevant to this study) live longer than men, although the gap between the sexes has decreased over time down to 3.7 years (*Sanders, 2017*).

We found that, in both datasets, burden of ultra-rare ($MAF<0.0001$) PTVs was negatively and significantly correlated with lifespan, and with healthspan in UKB (*Figure 1*). The proportional hazards effect estimations (sign and order of magnitude) were consistent, $\beta = 0.046$ and $0.014$ per mutation, for lifespan and healthspan in UKB, respectively. To estimate the effect of ultra-rare PTVs on lifespan and healthspan in years, we equate the contributions to log-hazards from the Gompertz term, $\Gamma = 0.093$ per year, and the burden term, $\beta$ per mutation: each additional ultra-rare PTV accounts for $\beta/\Gamma$ years of reduction, that is roughly 0.5 and 0.16 years per mutation for lifespan and healthspan, respectively. Moreover, the Cox regression coefficients were very similar in UKBBN and UKB datasets, indicating consistency of the effect across populations despite differences in population structure and morbidity statistics (*Table 2*), tissue source (blood in UKB and brain in UKBBN), sequencing methods and variant calling pipelines (*Keogh et al., 2017*).

We also observed a smaller but still significant effect of the ultra-rare PTV burden on mothers' but not on fathers' age at death in UKB. The effect size on mother's age at death was smaller and less significant than that on the subject's healthspan and lifespan.

The difference between male and female longevity is one of the most conserved observations in human biology. In light of this, we ran analysis separately for men and women and found sex-specific effects for lifespan phenotypes (*Table 3*). Association with age at death was similar between the sexes. However, the associations with healthspan and mother's age at death were almost entirely driven by women.

On average, we identified 6 ($SD = 2.6$) ultra-rare PTVs per genome (*Figure 1—figure supplement 1*, upper left corner). The variability of the burden of $SD = 2.6$ transforms into the variability in life- and healthspan reduction of 1.3 and 0.4 years, respectively. To visualize the effects of such PTVs on survival, we split deceased UKB subjects into five nearly equal groups corresponding to increasing PTV burden. The subjects in the first group had 0–3 ultra-rare PTVs per genome, in the second - 4 or 5 PTVs, in the third - 6 or 7 PTVs, in the forth - 8 or 9 PTVs and in the fifth - 10 or more PTVs (*Figure 2*, inset). Mean ages within the groups were 57.7, 57.5, 57.5, 57.4, and 57.4 years, respectively, with no difference in age distribution across the groups (Kolmogorov-Smirnov test on two samples p-value > 1%).

The Kaplan-Mayer (KM) survival curves for UKB individuals who harbor the lowest and the highest number of ultra-rare PTVs are shown in *Figure 2* as a function of the follow-up time. The separation of the curves further illustrated elevated mortality of the subjects with high PTV burden, with the

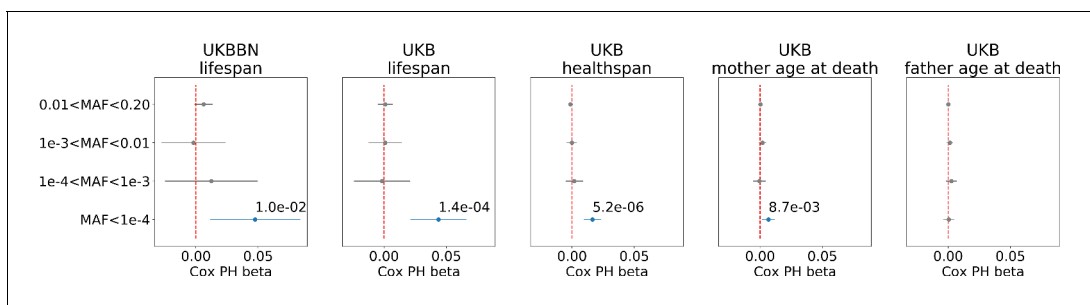

**Figure 1.** Association of burden of PTVs binned by population frequency (or minor allele frequency, MAF) with lifespan, healthspan, and parental age at death. Number of ultra-rare variants belonging to each MAF bin was calculated for each exome and tested for association with lifespan phenotypes using Cox proportional hazards model and covariates to account for population structure. UKBBN lifespan was tested for associations with corresponding PTVs burdens using sex and 20 first principal components (PCs) from PCA analysis with 1000G project. UKB lifespan during follow-up was tested for associations using sex, age of enrollment, assessment centers and 40 PCs provided by UKB as covariates. UKB healthspan, mother's and father's ages at death were tested for associations using sex, assessment centers and 40 PCs as covariates. Beta coefficients estimated by Cox proportional hazards model (Cox PH beta) are plotted as dots with whiskers representing 95% confidence intervals. p-Values are shown for significant results only. Blue color designates statistically significant associations. Red dashed line designates zero Cox PH beta coefficient value. MAF - minor allele frequency, PTV - protein-truncating variants (defined as stop codon gains, frameshifts, canonical splice acceptor/donor sites variant), UKB - UK Biobank, UKBBN - UK Brain Bank Network.

The online version of this article includes the following source data and figure supplement(s) for figure 1:

**Source data 1.** Source data for *Figure 1*.

**Figure supplement 1.** Distributions of PTV number per UKB exome depending on the variant population frequency (or minor allele frequency, MAF).

**Table 3.** Association of burden of ultra-rare (*MAF*<0.0001) PTVs with healthspan and mother's age at death is sex-specific. Number of ultra-rare variants was calculated for each genome and tested for association with lifespan phenotypes using Cox proportional hazards model and covariates to account for population structure (see Materials and methods). Beta coefficients reported in the 'coef' column. Bold font designates statistically significant associations. N - number of individuals analyzed, events - number of corresponding events reported in UK Biobank.

| Phenotype | Sex | Coef | Ci (2.5%) | Ci (97.5%) | p-value | N | Events |
|---|---|---|---|---|---|---|---|
| Death | female | 0.048 | 0.012 | 0.083 | 0.008 | 21742 | 450 |
| Death | male | 0.041 | 0.011 | 0.070 | 0.007 | 18626 | 672 |
| Mother age at death | female | 0.008 | 0.001 | 0.015 | 0.026 | 21320 | 12370 |
| Mother age at death | male | 0.006 | −0.002 | 0.013 | 0.130 | 17989 | 11081 |
| Father age at death | female | 0.002 | −0.004 | 0.008 | 0.558 | 20914 | 15679 |
| Father age at death | male | −0.001 | −0.008 | 0.006 | 0.796 | 17783 | 13785 |
| Healthspan | female | 0.024 | 0.014 | 0.034 | 4.1E-06 | 21742 | 5667 |
| Healthspan | male | 0.009 | −0.001 | 0.019 | 0.070 | 18626 | 6037 |

most significant difference between cohorts #1 and #5 (log-rank test $p = 7.1 \times 10^{-5}$). Due to Gompertz mortality acceleration, most of the death events involve the oldest individuals. Accordingly, the KM analysis here is naturally limited to a relatively narrow age group representing those close to the maximum age in the UKB population.

Having established the association of PTVs number with lifespan, we explored other types of genetic variants selected for incidence frequency and category: 3-prime and 5-prime UTR region variants, transcription factor (TF) binding sites, and structural interaction variants (*Table 4*). Among all tested PTV types, the most significant associations with lifespan and healthspan were observed for the ultra-rare (*MAF*<0.0001) stop gain, splice donor, and frameshift variants (*Figure 3*). However, only stop gains were associated with mother's age at death, and none of the categories were associated with father's age at death. As a negative control, we also included the effects of neutral variants - synonymous variants, which showed no significant associations with lifespan phenotypes.

## Gene constraints analysis

Ultra-rare PTVs affect 89% of the sequenced genes in the UKB dataset. No ultra-rare PTVs were observed in the remaining 11%, which we refer as genes intolerant to rare PTVs (iPTV). We compared these genes with those harboring at least one PTV ($n = 16,495$) within the same 4 MAF bins tested for the association with lifespan. iPTV genes, on average, were expressed in a higher number of tissues (*Figure 4a*) and had higher indispensability scores (a metric to measure gene essentiality introduced by *Khurana et al., 2013*; *Figure 4b*). As expected, iPTV genes in the UKB cohort are strongly enriched in genes intolerant to PTVs measured by pLI scores (*Figure 4—figure supplement 1b*), confirming that genes intolerant to PTVs largely overlap between UKB and ExAC datasets. Accordingly, genes that harbored frequent PTVs had tissue-specific expressions and had lower indispensability scores, thus were less essential, in agreement with previously published results for ExAC cohort (*Lek et al., 2016*). Ultra-rare stop gains were more likely to trigger nonsense-mediated mRNA decay (NMD) (*Figure 4c*) as predicted by snpEff by 50 base-pair rule (*Maquat, 2004*) which was also demonstrated for the rare variants in GTEx dataset (*Li et al., 2017*).

We further hypothesized that subjects with the same number of ultra-rare PTVs may have different lifespan due to difference in the damaging effect of their PTVs. Thus, subjects dying earlier would harbor more deleterious alleles than those dying later. To test this idea, we compared characteristics of genes disrupted by PTVs in subjects with the same PTV number (5 PTVs per exome, $n = 171$) but different lifespan (*Figure 5a*). Our analysis confirmed the idea that subjects dying younger harbored more damaging PTVs. Those variants affect more broadly expressed genes, based on GTEx gene expression data (*Figure 5c*), and cause gene loss-of-function more frequently (*Figure 5e*). PTVs in shorter-lived subjects also resided in genes less likely to maintain a wild-type phenotype when a single copy of the gene is inactivated, as evident by genome-wide haploinsufficiency score GHIS from *Steinberg et al. (2015)*; *Figure 5b*. Also, these PTVs affected genes with

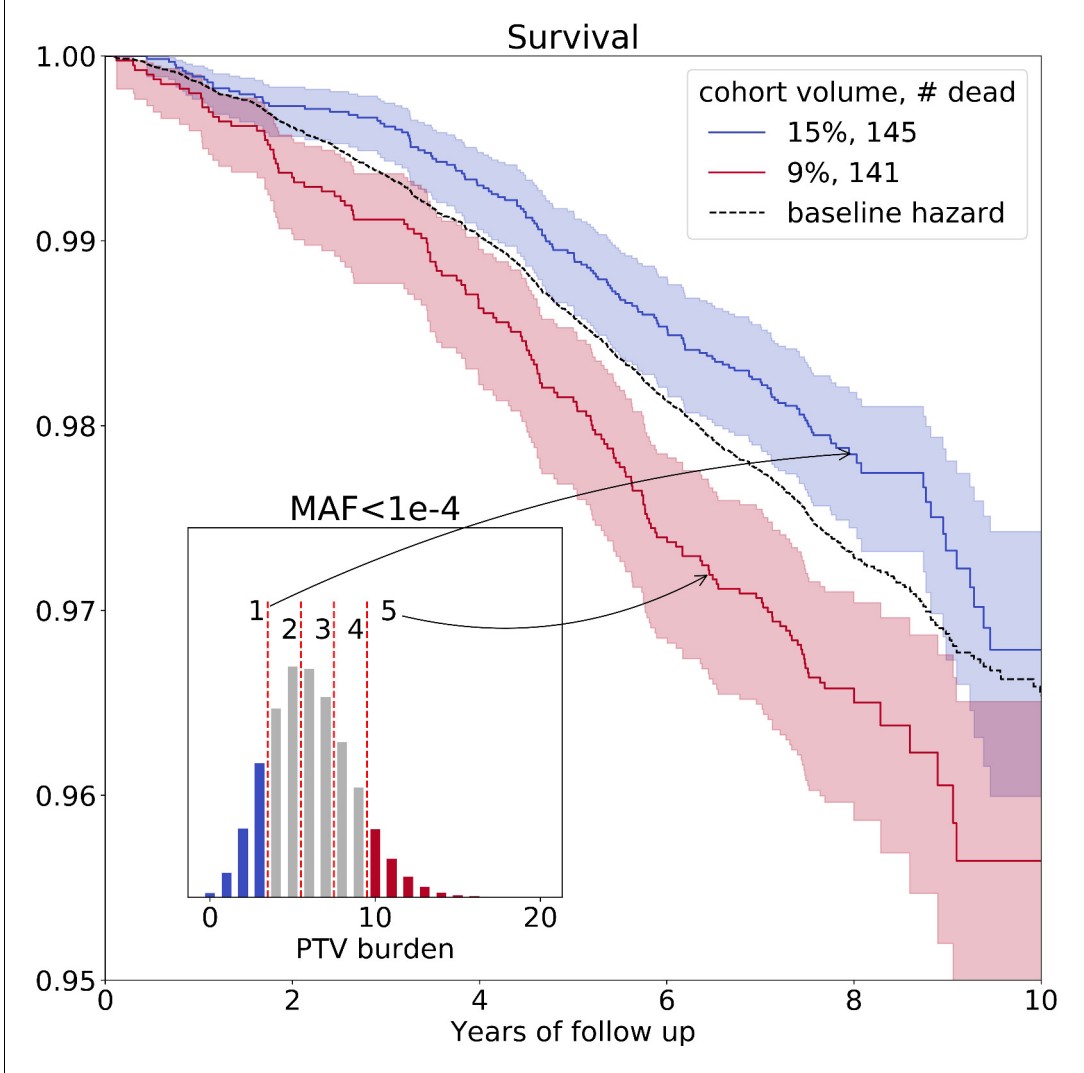

**Figure 2.** Ultra-rare (*MAF*<0.0001) PTV burden distribution and survival curves for the deceased UKB subjects stratified into groups based on the increasing burden. Blue line represents survival of individuals with low PTV burden (3 or less ultra-rare PTVs per genome) and red line represents survival of individuals with high PTV burden (10 or more ultra-rare PTVs per genome) during eleven years of follow-up (log-rank test $p = 7.1 \times 10^{-5}$). The absolute number of deceased subjects in each line, and the corresponding percentage, is indicated in the legend. The inset shows the distribution of the number of ultra-rare (*MAF*<0.0001) PTVs per deceased individual in UKB cohort, colored accordingly to the survival curves. MAF - minor allele frequency, PTV - protein-truncating variants (defined as stop codon gains, frameshifts, canonical splice acceptor/donor sites variant).

stronger constraints against PTVs, based on the observed/expected LoF (oe, gnomAD v2.1) scores, suggesting that these genes harbored fewer PTVs than expected in the general population. Mean values for constraints of the genes disrupted by PTVs showed association across lifespan tested by Cox PH model (*Figure 5a*, inset). Percent of tissues expressing these genes, oe scores and loss-of-function occurrence were all significantly associated with lifespan of subjects with 5 PTVs (*Figure 5a*, inset).

## Gene burden test

We further tested whether some of the genes were more frequently affected by ultra-rare PTVs in UKB individuals depending on their lifespan and healthspan. For that, we 1) split the list of UKB subjects ordered by lifespan or healthspan into two groups of same size, 2) summed up a number of unique cases of the gene harboring ultra-rare PTV for each group, and 3) tested whether those numbers are biased towards one of the groups by Fisher's exact test. Interestingly, none of the genes

**Table 4.** Variant annotations for 8,959,608 SNPs from FE dataset which is part of UKB.
Variant types selected for analysis are written in italics, and PTV burden components marked in bold.
Some variants may have multiple effects. PTV - protein-truncating variants, UTR - untranslated region,
TF - transcription factor, TFBS - transcription factor binding site.

| Variant effect | Number of variants |
| --- | --- |
| intron_variant | 3643472 |
| missense_variant | 2281322 |
| synonymous_variant | 1159078 |
| splice_region_variant | 333226 |
| downstream_gene_variant | 329399 |
| upstream_gene_variant | 303346 |
| *3_prime_UTR_variant* | 303346 |
| *5_prime_UTR_variant* | 192159 |
| ***frameshift_variant*** | 96359 |
| intragenic_variant | 85619 |
| sequence_feature | 79868 |
| ***stop_gained*** | 68054 |
| *structural_interaction_variant* | 57365 |
| *TF_binding_site_variant* | 45909 |
| *5_prime_UTR_premature_start_codon_gain_variant* | 34381 |
| ***splice_donor_variant*** | 22476 |
| *disruptive_inframe_deletion* | 21392 |
| ***splice_acceptor_variant*** | 18591 |

*Table 4 continued on next page*

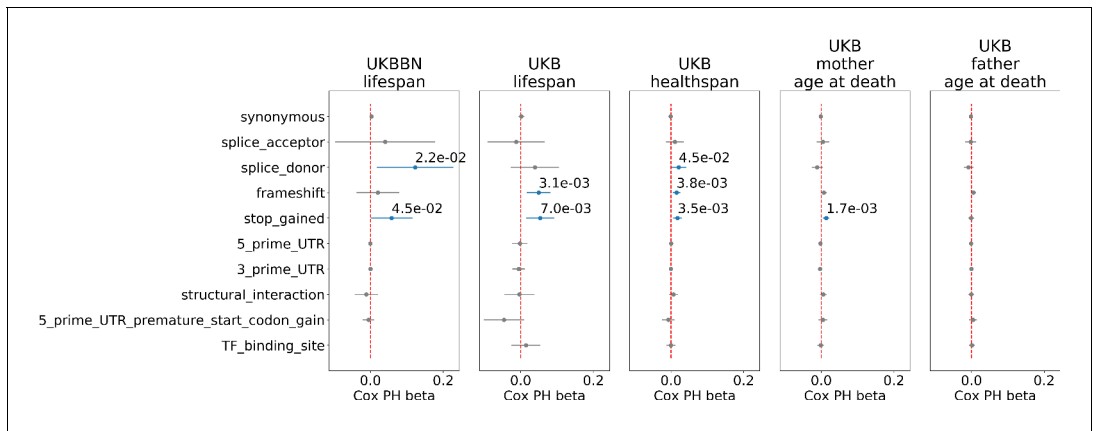

**Figure 3.** Association of ultra-rare ($MAF$<0.0001) variants burden with UKB and UKBBN lifespan, UKB healthspan, and parental longevity (father's and mother's age at death). The number of ultra-rare variants belonging to each category was calculated for each genome and tested for association with lifespan phenotypes using Cox proportional hazards model and covariates to account for population structure. UKBBN lifespan was tested using sex and 20 first principal components (PCs) taken from principal component analysis of common variants shared between UKBBN and 1000G project. UKB lifespan during follow-up was tested for association with ultra-rare variants burdens using sex, age of enrollment, assessment centers, and 40 PCs provided by UKB as covariates. Sex, assessment centers, and 40 PCs were used as covariates for associations with UKB healthspan, and mother's and father's age at death. Beta coefficients estimated by Cox proportional hazards model (Cox PH beta) are plotted as dots with whiskers representing 95% confidence intervals. p-Values are shown for significant results only. Blue color designates statistically significant associations. Red dashed line designates zero Cox PH beta coefficient value. UKB - UK Biobank, UKBBN - UK Brain Bank Network, TF - transcription factor, UTR - untranslated region, MAF - minor allele frequency, PTV - protein-truncating variants (defined as stop codon gains, frameshifts, canonical splice acceptor/donor sites).
The online version of this article includes the following source data for figure 3:

**Source data 1.** Source data for *Figure 3*.

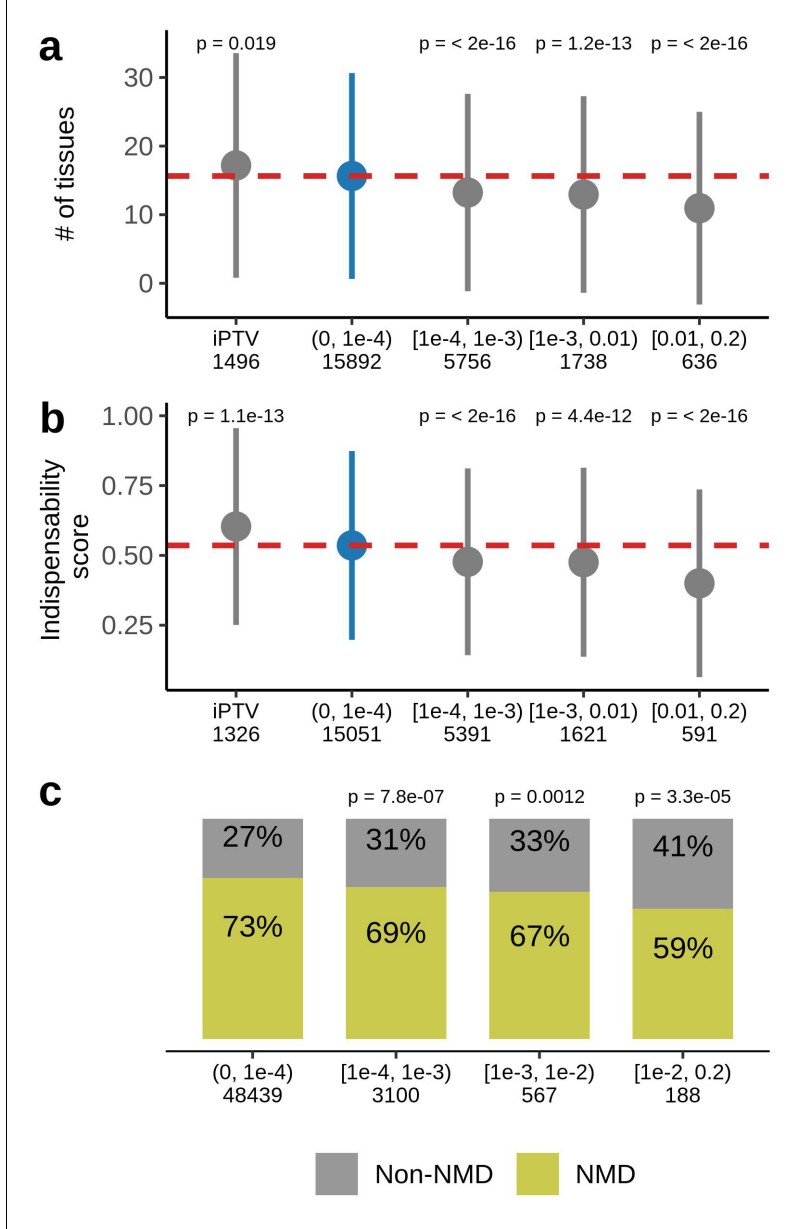

**Figure 4.** Characteristics of genes harboring PTVs binned by allele frequency in UKB. (**a**) PTV-intolerant (iPTV) genes and genes harboring ultra-rare PTVs ([$0, 1e - 4$) bin) are more broadly expressed and (**b**) have higher indispensability scores (a metric to measure gene essentiality introduced by *Khurana et al., 2013*). The results of comparisons are grouped in subsequent MAF bins and the numbers in the horizontal axis represent the number of genes included in the analysis. (**c**) Ultra-rare stop gains are more likely to trigger nonsense-mediated decay (NMD) based on 50 bp rule prediction, the numbers in the horizontal axis represent the total number of stop gains in each bin. Each group was compared to the bin of rarest variants [$0, 1e - 4$), where PTVs are significantly associated with lifespan. p-Values in (**a**) and (**b**) are calculated using Wilcoxon rank-sum test, p-values in (**c**) are calculated using Fisher's exact test. NMD - nonsense-mediated mRNA decay, PTV - protein-truncating variant (defined as stop codon gains, frameshifts, canonical splice acceptor/donor sites variant), iPTV - genes intolerant to PTV. The online version of this article includes the following figure supplement(s) for figure 4:

**Figure supplement 1.** Comparison of the constraints of genes that harbors protein-truncating variants with different MAFs and genes free of those variants.

**Figure supplement 2.** Distribution of ultra-rare PTVs across human genome normalized by number of total variants.

*Figure 4 continued on next page*

*Figure 4 continued*

**Figure supplement 3.** Relationship between the number of ultra-rare protein-truncating variants and the odds ratio obtained in the Fisher's exact test for each gene.
**Figure supplement 4.** Distribution of oe scores in genes with odds ratio (OR) below 1 (OR <1), OR greater than 1 (OR >1), and the rest of the genes which obtained a non-significant p-value (>0.05).

reached statistical significance under FDR p-value cut-off of 0.05 (*Supplementary file 1* and *Supplementary file 2*).

Previous efforts to characterize rare PTVs in large datasets demonstrated that some genes are more prone to be affected by rare PTVs than others (*Lek et al., 2016*). However, it was unclear to what extent that applies to the UKB dataset. We first assessed the genome-wide distribution of ultra-rare PTVs and found it to be similar to the rest of the variants (*Figure 4—figure supplement 2*). To address the same question at the level of individual genes, we estimated the number of ultra-rare stop gains and frameshifts per gene in the UKB cohort, as well as the number of all synonymous variants per gene as a neutral read-out to normalize for coverage. By comparing the number of PTVs and synonymous variants per gene, to the overall number of those variants, we found 110 genes that were prone to ultra-rare PTVs, as well as 188 genes intolerant to those variants (*Supplementary file 3*, *Figure 4—figure supplement 3*). Our estimations were in agreement with oe scores provided by gnomAD. As expected, genes prone to ultra-rare PTVs in UKB showed high oe scores in gnomAD, while genes intolerant to those variants had low oe scores (*Figure 4—figure supplement 4*). We excluded variants in splice donor and acceptor sites, as the number of variants per gene is directly affected by the number of introns.

## Somatic mutations and mortality acceleration

In addition to the germline burden of extremely rare PTVs, somatic cells accumulate new genetic variants (*Vijg, 2000*; *Milholland et al., 2015*; *Zhang and Vijg, 2018*) at a median mutation frequency of $R \approx 10^{-8}$ per base pair per year (*Milholland et al., 2017*). Thus, the negative effects on healthspan and lifespan due to germline burden should get gradually amplified with age in somatic cells. We quantitatively assessed whether the contribution of accumulated somatic PTVs is strong enough to explain the exponential growth of mortality with age, a.k.a. the Gompertz law. In doing so, we extrapolated the Cox PH model for germline PTV burden by taking into account the effects of acquiring new PTVs with age in somatic cells. The somatic PTV burden increases linearly with age and can be estimated as $\lambda L R t$, where $t$ is age, the genome size is $L = 3$ Gbp, and the fraction of the genome covered with the extremely rare PTVs (10 kbp) with $MAF < 10^{-4}$ is $\lambda = 0.33 \cdot 10^{-5}$. Overall, the somatic PTV burden contributed to the mortality log-hazard a linear (in age) term $\beta \lambda L R t$, where $\beta = 0.046$ per year is the Cox PH coefficient, whereas the Gompertz contribution would be proportional to $\Gamma t$. We estimate that the somatic PTV burden term $\beta \lambda L R \approx 4.6 \cdot 10^{-6}$ per year is negligible in comparison to the Gompertz exponent $\Gamma \approx 0.09$ per year characterizing mortality and incidence of chronic disease acceleration with age (*Zenin et al., 2019*). The estimated effect of somatic PTV accumulation is minor in comparison to the effect of germline PTV burden, and can account only for a minute fraction of mortality and morbidity acceleration. However, our prediction should be experimentally validated, for example, by testing the effect of somatic mutations on lifespan and healthspan in model organisms.

## Discussion

We report that both lifespan and healthspan are negatively impacted by the burden of rarest variants that disrupt genes and are already present at birth. These mutations are found in most human subjects. Thus, disease occurrence and age of death are directly influenced by genotype. Our approach is radically different from the previous searches for variants that contribute to longevity, for example searches for alleles enriched in centenarians. In this regard, we find that the genetic component of lifespan is shaped by two mutation types: common variants, as found by previous studies, and ultra-rare PTVs, as shown here.

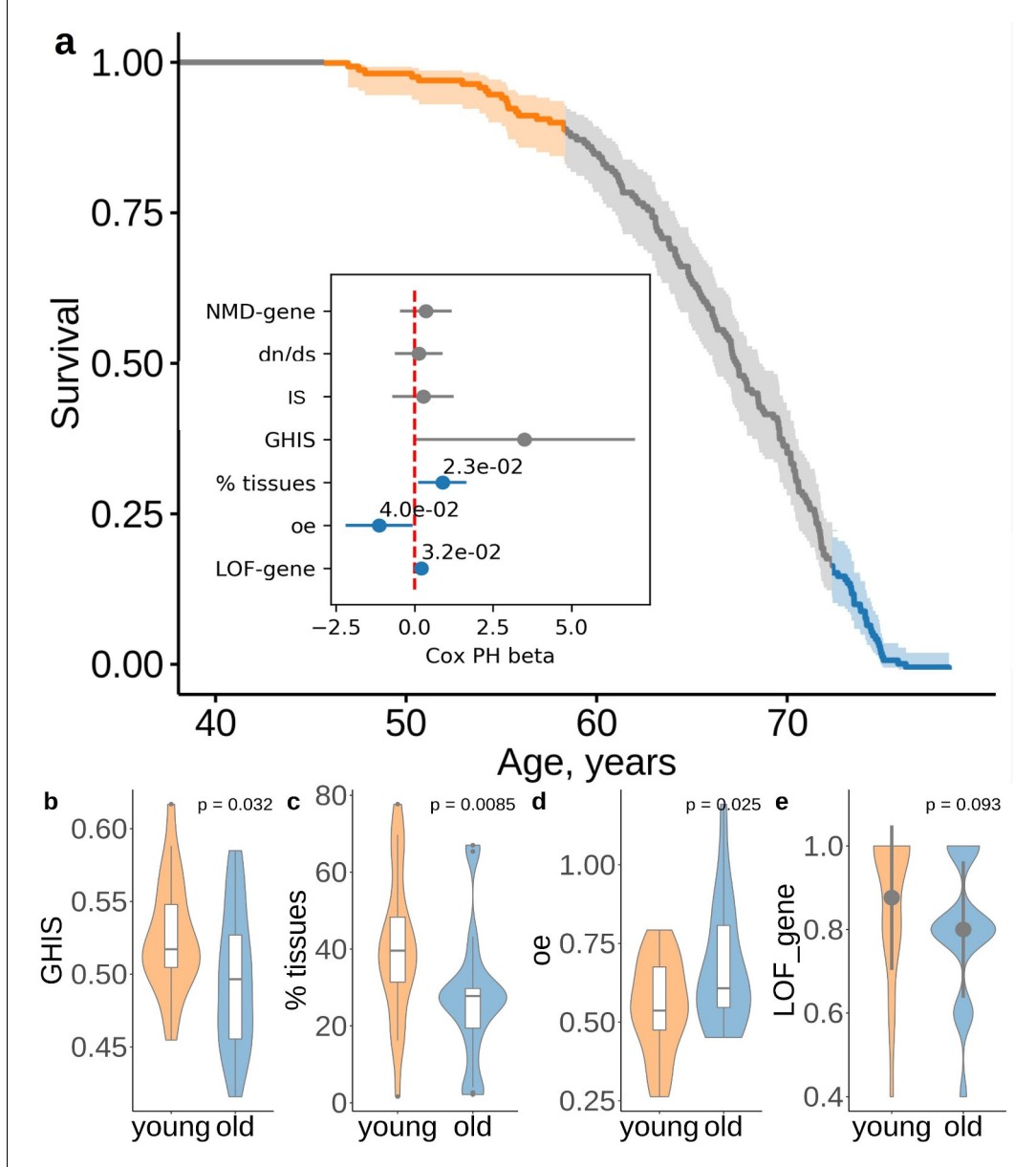

**Figure 5.** Deleterious effect of the ultra-rare PTVs are also associated with lifespan. (a) Survival of UKB subjects with 5 ultra-rare PTVs per exome. The inset shows association between lifespan and the properties of genes harboring ultra-rare PTV: evolutionary constraint quantified by $dN/dS$ ratios (the ratio of substitution rates at non-synonymous and synonymous sites) in human-chimpanzee orthologs; indispensability score (IS) as in *Khurana et al., 2013*; genome-wide haploinsufficiency score (GHIS) as in *Steinberg et al. (2015)*; (relative) number of tissues expressing the gene; observed/expected (oe) score; prediction for variants being loss-of-function (LOF, see LOF-gene) and triggering NMD (see NMD-gene). Orange and blue areas in (a) designate survival windows for subjects dying earlier in life (young) and later in life (old) and this color scheme is the same as that in the plots B-D. Difference in (B) GHIS scores, (C) percent of tissues expressing gene affected by variants, and (D) oe scores, and (e) proportion of predicted loss-of-function (*LOF_gene*) variants for individuals with same PTV number but differing in lifespan (i.e. dying younger ($47.4 - 58.9$ years) or older ($73.8 - 78.5$ years)). p-Values in (b) and (d) were calculated by Student t-test, p-value in (c) and (e) were calculated by Wilcoxon rank-sum test. NMD - nonsense-mediated decay, IS - indispensability score, GHIS - genome-wide haploinsufficiency score, LOF - loss of function, PTV - protein-truncating variant (defined as stop codon gains, frameshifts, canonical splice acceptor/donor sites variant).

These conclusions are based on the analysis of two large independent datasets that provide both whole exome sequencing data and lifespan traits. Previously released genotyping data were probed primarily for common variants, thus missing information on the most deleterious, rare variants. Here,

we took advantage of the recent release of UKB exomes and revealed the relationship between the most damaging variants, ultra-rare PTVs, and lifespan traits.

However, due to the limited follow-up, mortality in the UKB dataset reflects the progression rate of age-related chronic diseases in an individual; that is if a subject is deceased, he/she most probably had one or more age-related disease at the time of enrollment. As a result, UKB subjects included in lifespan analysis are biased toward shorter lifespan. A general UK population born at the same time had a life expectancy at birth of 71 years which is much longer compared to the average lifespan of 57 years in the deceased UKB cohort. On the other hand, UKBBN subjects had an average lifespan of 69 years which is closer to the actual lifespan in the UK. However, subjects in the UKBBN cohort were born in the period spanning the whole century from 1891 to 1996. As one would expect, deleterious alleles might accumulate in recent generations due to advances in medicine. Thus, our UKBBN analysis of a mixture of a few generations could be biased. Both limitations can be addressed 30–40 years later when the average lifespan of UKB individuals will reach the average lifespan of the UK population.

The association of ultra-rare PTV mutations with healthspan, however, reflects the effect of deleterious gene variants on the incidence of the first chronic disease and thus covers the accumulated effect of genotype on health and survival over a much longer time, effectively from birth up to the age of enrollment/death.

The association of ultra-rare mutation burden and lifespan is consistent between the UKB and UKBBN cohorts. UKB has more subjects but a narrower age distribution, and UKBBN provides post-mortem genotypes of individuals deceased at ages of $16 - 105$ years old. By the nature of its design, the UKBBN cohort may be enriched for individuals prone to diseases and death at any given age. On the other hand, UKB subjects exhibit lower mortality and hence are probably healthier than the general population (*Ganna and Ingelsson, 2015*). It, therefore, appears that the association of ultra-rare mutations and lifespan traits is a general feature that applies to the UK population. However, there is still an open question whether our findings are translatable to the populations outside UK and Europe. The release of more ethnically diverse datasets accompanied by lifespan phenotypes would help to address this question in the future. It is also unclear if the association would be preserved beyond the 11 years of the follow-up period in UKB. Findings from a much older UKBBN cohort suggest that the effect size of ultra-rare PTVs on lifespan will remain significant. To fully understand the role of rare variants in human lifespan, we need to test its effects in an older UKB cohort as well as in large ethnically diverse datasets.

We found no association between father's age at death and PTV burden, while there was a modest effect for mother's age at death. At the same time, a genetic correlation between longevity and father's age at death was reported to be one of the strongest (*Deelen et al., 2019*). One of explanations would be that ultra-rare PTV burden is more relevant for lifespan than for longevity. To address this hypothesis in the future, PTV burden can be compared between centenarians and an appropriate control group. If PTV burden is associated with longevity, we would expect centenarians to be severely depleted of ultra-rare PTVs.

Interestingly, we observed sex-specific effects of PTV burden for healthspan and mother's age at death. Both signals were mostly driven by women. At the same time, men had a much shorter healthspan compared to women (Cox ph beta = 0.16, p-value=1.55E-18). We hypothesize that the genetic component, represented here by ultra-rare PTVs, may play a less important role in male healthspan due to the lifestyle choices such as smoking, drinking, risky behavior and unhealthy diet. Indeed, men are known to smoke more (*Peters et al., 2014*), drink more, and exhibit higher BMI scores (*Wills et al., 2017*) than women in UKB. To investigate sex specificity in more detail, we ran an analysis of the X-chromosome in men, where mutations cannot be compensated by in the homologous chromosome. We found no associations of X-chromosome PTV burden with either lifespan or healthspan. However, the number of PTVs per individual specifically on X chromosome is extremely low; thus, we may be out of power to pick up the difference.

Ultra-rare PTVs occur across the genome and affect 89% of sequenced genes in UKB. Intriguingly, we observed a subset of 1,496 genes that are free of ultra-rare PTVs in the whole UKB population sequenced so far. These genes are more essential as evidenced by high indispensability scores and are expressed more broadly throughout the body. Together, this findings indicate strong purifying selection against PTVs in these genes. Their disruption could lead to either childhood or embryonic

lethality, the time periods that are not covered by UKB as well as other public datasets, for example ExAC.

As expected, genes affected by rare and common PTVs ($MAF$>0.0001) are less evolutionary conserved and more frequently disrupted in the general population (*Figure 4—figure supplement 1*), less essential (based on indispensability scores), and expressed in fewer tissues (*Figure 4a,b*). Moreover, fewer common nonsense variants were predicted to trigger nonsense-mediated mRNA decay; thus, they affect gene expression less than ultra-rare stop gains (*Figure 4c*). Overall, common PTVs are expected to have a lower effect on fitness, which would explain the lack of the association between the burden of common PTVs and lifespan phenotypes. It is apparent that the ultra-rare PTVs are more damaging than common PTVs but not enough damaging to cause early life mortality.

Notably, individuals sharing same PTV number still have diverse lifespan. This discrepancy might be explained by differences in the rate of age-related damage accumulation, regulated by environment and genetic factors. In addition, the impact of a PTV on phenotype depends on the gene it disrupts and its position within the gene body. For example, disruption of a more evolutionary conserved gene and with a broad expression would, intuitively, have a stronger effect on lifespan than the disruption of a less conserved gene with a tissue-specific expression. Indeed, our analysis confirmed that individuals with shorter lifespan were born with more deleterious alleles. Genes disrupted in short-lived individuals are broadly expressed in the body, are more likely to cause haploinsufficiency when inactivated, and are more intolerant to PTVs (according to gnomAD oe scores). Additionally, PTVs in subjects with shorter lifespan were more likely to cause gene loss-of-function. Thus, both the degree of damage caused by ultra-rare PTVs and the number of these variants are important factors influencing human lifespan.

Intriguingly, the effect size of ultra-rare PTVs on lifespan was comparable to the effect of known longevity alleles. For example, $\epsilon4$ allele in *APOE/TOMM40* locus conferred an estimated 1.24 years of life shortening in women, as inferred from a large parental survival study (*Joshi et al., 2016*). The PTV burden difference of $2-3$ variants corresponds to the similar effect size ($1-1.5$ years) on the lifespan variation at the standard deviation for $MAF$<0.0001.

Having established the mortality and morbidity risk association with PTVs, we were able to factor in the rates of somatic mutation accumulation over the lifespan. The dramatic discrepancy between the estimate for somatic PTV burden accumulation and the empirical mortality and morbidity acceleration does not support the hypothesis that random somatic mutations significantly reduce healthspan or lifespan. Moreover, the analysis shows that the effect of accumulation of somatic mutations is less profound than that of germline PTV burden. Thus, we found little evidence for a significant role of somatic mutations in aging (*Promislow and Tatar, 1998*; *Moorad and Promislow, 2008*). Somatic mutations may, however, play a role through high-order effects, such as clonal expansion and cell competition, and hence amplify the effects of other forms of damage (*Martincorena, 2019*).

Taken together, the effects of common variants earlier implicated in longevity and the effects of ultra-rare variants reported here could help explain the apparent heritability of lifespan. Currently, this issue is not fully resolved. Twin studies (*Herskind et al., 1996*; *Ljungquist et al., 1998*) suggest that lifespan could be as much as $23-33\%$ genetically determined. A more recent study (*Ruby et al., 2018*) puts up a challenge to this conclusion and points to a much lower level of genetic determinism. We therefore expect that future investigations of the effects of ultra-rare genetic variants may turn to be crucial for quantitative understanding of lifespan heritability.

These findings strengthen the case for complexity of aging, wherein aging is a systemic process resulting from the combined accumulation of age-related deleterious changes, none of which could cause aging on their own (*Gladyshev, 2016*). The advantage of mutations in aging studies, however, is that they can be quantified and their contribution estimated, which is something that is currently much more difficult to do for other forms of age-related damage.

## Materials and methods

### UKB cohort

The first batch of UKB exome sequence group consists of 49,960 individuals who passed QC procedures by UKB. Exome sequencing cohort is enriched with samples with a higher rate of imaging and enhanced measurements such as retinal optical coherence tomography test, visual acuity, hearing

test, and other. This cohort is not biased on any health condition, disease or physical measurement results from the UKB population of almost 500,000 individuals (*Hout et al., 2019*). We selected a cohort of 41,250 individuals who self-reported 'White British' and have very similar genetic ancestry based on a principal components analysis of the genotypes. Then, we made an effort to produce the maximal independent set of individuals based on computed kinship coefficients (two individuals were considered related if they share relatedness of third degree or closer) and selected 40,368 individuals for the analysis.

## Exome data

Exome data consisted of 8,959,608 SNPs and short indels from human coding DNA. We selected 6,208,943 variants that are not monomorphic in UKB cohort and have a missing rate less than 10% and $MAF<0.2$. We annotated these genetic variants for functional consequence using SNPeff (*Cingolani et al., 2012*) software and *GRCh38.86* genome reference. UKBBN dataset was additionally annotated with ANNOVAR (*Wang et al., 2010*) to add ExAC MAFs. In downstream analysis we focused on protein-truncating variants annotated as: stop codon gained, frameshift variant, slice donor or splice acceptor site, this produced 152,790 and 11,393 SNPs and indels in UKB and UKBBN, correspondingly.

## PTV burden calculation and Cox proportional hazards model

PTV burden was defined as a number of ultra-rare ($MAF<0.0001$) variants that disrupt open reading frame (stop gain, frameshift, disruption of splice donor/acceptor site). PTV burden was tested for association with UKBBN lifespan using Cox PH model with sex and first 20 principal components (obtained by clustering with 1000G dataset, see below) as covariates in R (*R Development Core Team, 2018*). For UKB data we included sex, 40 genetic principal components and assessment centers as covariates for Cox PH analysis on lifespan, healthspan and mother's and father's age at death. For all types of survival data except for healthspan we have also added age at assessment as covariate. Genetic principal components were calculated on genotypes for 500,000 UKB participants (*Bycroft et al., 2018*).

## UKBBN PCA with 1000G

First and second chromosome for all 1000G super populations and UKBBN dataset were clustered together. For that, 1000G vcf files were lifted over to hg19 using picard tools (*Broad Institute, 2018*) combined with UKBBN vcf file by overlap variants using GATK tools (*Van der Auwera et al., 2013*). Variants with MAF deviating between datasets over 30% were excluded. Eigen vectors were obtained from variants with $MAF>10\%$ pruned using 50 window size, step size of 5 and variance inflation factor threshold of 1.5 by Plink (*Purcell et al., 2007*). We kept individuals that clustered with EUR superpopulation.

## Data filtering

PTVs in UKB were filtered using internal MAFs. Since UKBBN cohort is much smaller to get desired resolution we used ExAC MAFs for non-finish European population (`ExAC_NFE`). We excluded ultra-rare variants absent in ExAC dataset (`ExAC_ALL = 0`) from UKBBN analysis to reduce number of sequencing and variant calling errors. Analysis in both datasets was restricted to autosomal chromosomes to avoid sex bias. We restricted UKBBN cohort to natural causes of death (i.e. excluding car accidents, poisoning and suicides) and excluded deaths with no abnormalities detected.

## Data sources

UKBBN vcf files were downloaded from EGA repository (EGAS00001001599, https://www.ebi.ac.uk/ega/studies/EGAS00001001599). Transcripts per kilobase million (TPM) counts for 53 human tissues were downloaded from GTEx Portal, release v7. Gene expression values within brain regions, two heart and two skin samples were averaged for subsequent analysis, and primary cell cultures were excluded, yielding a total of 37 tissues. Transcripts considered to be expressed in the tissue if $TPM>10$. Oe ratios were downloaded from gnomAD repository (`gnomad.v2.1.1.lof_metrics.by_gene.txt.bgz`). GHIS values were obtained from *Steinberg et al. (2015)* and indispensability scores were downloaded from *Khurana et al., 2013*. *dS* and *dN* values for chimpanzee-human

orthologs were downloaded from Ensembl Biomart. NMD and LoF predictions were obtained from snpEff annotation ('NMD.gene', 'LoF.gene') (*Cingolani et al., 2012*). All UK Biobank data are available upon application.

## Gene burden analysis

Gene burden analysis was performed with assumptions that all ultra-rare PTVs would have the same effect direction and the same effect size. Following those assumptions, we summed up all cases of gene harboring ultra-rare PTV. Cohorts were defined by splitting UKB into two groups with equal number of subjects based on ordered lifespan or healthspan data. We tested the hypothesis that some genes harbor more ultra-rare PTVs in one cohort than another (compared to the sum of PTV number in each cohort) using Fisher's exact test. To explore sex-specific effects, we separately run analysis for healthspan in males and females. In order to identify genes with a significantly deviated burden of ultra-rare PTVs in UKB, we performed a Fisher's exact test using the number of ultra-rare PTVs and synonymous variants. For each gene, we build a 2 × 2 contingency table containing the number of ultra-rare PTVs observed in the gene and those observed in the rest of the population, and the number of synonymous variants observed in the gene and those observed in the rest of the population. The result of each test was an odds ratio and p-value, where genes with odds ratio <1 showed a disproportionately low number of rare PTVs. The Fisher test was performed using the fisher.test function in R, and the Bonferroni correction is performed using p.adjust function in R.

## Acknowledgements

This research has been conducted using the UK Biobank Resource under Application Number 21988. The study was funded by NIH (to VNG) and by Gero LLC. The provision of UKBBN data used in this study was supported by funding from the UK Medical Research Council and BDR (Brains for Dementia Research). The authors thank Prof. Dmitry Ivankov for a helpful discussion.

## Additional information

### Competing interests

Aleksandr A Zenin, Andrei E Tarkhov: Employed by Gero LLC. Peter O Fedichev: Founder of Gero LLC. The other authors declare that no competing interests exist.

### Funding

| Funder | Grant reference number | Author |
|---|---|---|
| National Institute on Aging | AG047745 | Vadim N Gladyshev |

The funders had no role in study design, data collection and interpretation, or the decision to submit the work for publication.

### Author contributions

Anastasia V Shindyapina, Conceptualization, Resources, Data curation, Formal analysis, Investigation, Visualization; Aleksandr A Zenin, Conceptualization, Resources, Data curation, Software, Formal analysis, Validation, Investigation, Visualization, Methodology; Andrei E Tarkhov, Resources, Data curation, Software, Formal analysis, Visualization; Didac Santesmasses, Data curation; Peter O Fedichev, Conceptualization, Supervision, Funding acquisition, Investigation, Project administration; Vadim N Gladyshev, Conceptualization, Supervision, Funding acquisition, Project administration

### Author ORCIDs

Anastasia V Shindyapina https://orcid.org/0000-0002-7336-3086
Aleksandr A Zenin https://orcid.org/0000-0003-1522-0359
Andrei E Tarkhov http://orcid.org/0000-0003-3350-4785
Vadim N Gladyshev https://orcid.org/0000-0002-0372-7016

## Ethics

Human subjects: Deidentified exome sequences were analyzed.

## Decision letter and Author response

Decision letter https://doi.org/10.7554/eLife.53449.sa1
Author response https://doi.org/10.7554/eLife.53449.sa2

# Additional files

## Supplementary files

• Supplementary file 1. Statistics from gene burden test for lifespan in UKB. Burdens of ultra-rare PTVs for each gene were compared between subjects with short and long lifespan.

• Supplementary file 2. Statistics from gene burden test for healthspan in UKB. Burdens of ultra-rare PTVs for each gene were compared between subjects with short and long healthspan in both sexes, and separately in females and males .

• Supplementary file 3. Statistics from gene burden test of ultra-rare PTVs in UKB population. Burden of ultra-rare PTVs for each gene and burden of synonymous variants was compared to the global burdens of ultra-rare PTVs and synonymous variants.

• Transparent reporting form

## Data availability

All data generated or analyzed during this study are included in the manuscript and supporting files. Source data files have been provided for Figures 1 and 3.

The following previously published dataset was used:

| Author(s) | Year | Dataset title | Dataset URL | Database and Identifier |
|---|---|---|---|---|
| Bycroft C, Freeman C, Petkova D, Band G, Elliott LT, Sharp K, Motyer A, Vukcevic D, Delaneau O, O'Connell J | 2018 | UK Biobank | https://www.ukbiobank.ac.uk | UK Bio Bank, NA |

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
