## [Decision Letter]

**Acceptance summary:**

We believe that your work will be of great interest to the aging research community. The paper will add to the current understanding of genetics of lifespan by demonstrating the importance of rare variants and PTVs.

**Decision letter after peer review:**

Thank you for submitting your article "Germline burden of rare damaging variants negatively affects human healthspan and lifespan" for consideration by *eLife*. Your article has been reviewed by three peer reviewers, and the evaluation has been overseen by Sara Hagg as Reviewing Editor and Jessica Tyler as the Senior Editor. The following individual involved in review of your submission has agreed to reveal their identity: Joris Deelen (Reviewer #3).

The reviewers have discussed the reviews with one another and the Reviewing Editor has drafted this decision to help you prepare a revised submission.

Summary:

The paper by Shindyapina and colleagues reports the results from a study on the cumulative effect of rare variants, identified using exome sequencing data, on healthspan and lifespan. The authors showed that a higher burden of ultra-rare protein-truncating variants is associated with increased lifespan and healthspan. Overall, the reviewers agreed on the fact that this is an interesting analysis using a unique dataset that will be an important contribution to the field of aging biology. Some suggestions may still improve the paper and are needed before it can be accepted for publication.

Essential revisions:

1) The difference between male and female longevity is one of the most conserved observations in human biology. In this light, it is important to see if any effects are sex specific or sex biases? Once the sexes are separated, is the effect observed in both or is it stronger in one sex? It would be interesting if the authors can speculate a bit more (in their Discussion section) about why they think they observed an association with mother's, but not father's age at death, given that the genetic correlation of longevity with father's age at death is much stronger than with mother's age at death (see Deelen et al., 2019). Hence, it may be that the ultra-rare PTV burden is relevant for lifespan, but not so much for longevity. A related but non overlapping question is whether PTVs have more damaging effects when on the X chromosome, where they are not compensated for by a second allele in males?

2) Are there genes with more frequent ultra-rare PTVs than other genes based on what is explored in the paper? This could be tested with gene-based burden tests, where more frequent PTV regions should be investigated in terms of functional annotation.

3) On the contrary, the ~1500 genes without PTVs, are they also found to be "human gene knock-outs"? What are the overlap of these identified genes with those identified in other similar projects?

4) More info on the Cox models is needed. What do the proportional hazards look like, Schoenfeldt residuals? Describe the underlying time scale: does it matter if it is age or time since measurement? Are there time-varying effects here, for example in the somatic mutation accumulations? Provide all estimates with 95% CI’s, less focus on p-values overall in the manuscript is desired. Related, in Figure 2, why did the authors decide to use follow-up time as timescale for their Kaplan-Meier curve? It would make more sense to use the actual age of the included individuals (as done in Figure 4A) given that the age range is quite narrow.

5) The section about somatic mutations and mortality acceleration is currently only based on estimations of somatic variation and computational modeling. It would be better if the authors use the actual sequencing data to try to detect somatic variants (using variant allele fractions) and subsequently associate these with lifespan and healthspan.

6) Are the analyses affected by the errors on mis-mapping encountered in the exome data from UKB? If so, have the authors re-ran the analyses after correcting errors?

7) Figure 4C; Did the authors only compare the 0, 1e^-4^ bin with the 1e^-4^, 1e^-3^ bin or was that the only comparison showing a significant difference? If the latter is the case, they should also report the P-values for the comparison with the other 2 bins.

8) Figure 5: The authors should add a panel with the difference in the LOF-gene between young and old (even if this is not significant), given that this one is also significantly associated with lifespan (Figure 5A).

9) Cohort differences should be acknowledged: are these cohorts representative of the general UK population? There is some discussion around this but more is needed. What are the implications if these cohorts are prone to selection bias? Authors should discuss the possibility that data beyond the 11 years of follow-up in UKB may lead to a bias in the results even if it's not possible to account for it directly. What are the mean ages at death in UK in general, and in the two cohorts?

10) The Introduction section currently contains some statements that are incorrect, and should be addressed:

"For example, GWAS on centenarians consistently demonstrate the loci near LPA and APOE, FOXO3A, HLA-DQA1 and SH2B3 genes to be associated with longevity (Serbezov et al., 2018)". The majority of the loci mentioned here have consistently been associated with parental lifespan, but not with longevity (which is defined as living to an age above a certain age threshold). The only locus consistently associated with longevity (including studies containing centenarians) is APOE. Hence, the authors need to update this statement and use a more appropriate reference here, such as the recent review from Melzer et al., 2019.

"However, the combined contribution of these variants could explain only a small part of lifespan heritability, at least as asserted from twin studies (Ruby et al., 2018)." The paper by Ruby et al. used pedigree data to estimate the heritability of lifespan. Hence, the authors cannot use this reference to refer to the heritability as estimated by twin studies and should thus update this reference or change their statement.

"We hypothesized that the rest could be explained by the combined burden of rare damaging gene variants". Why do the authors assume that the rare variants have to be damaging? It could well be that there is a contribution of variants that lead to increased functioning of genes/proteins and are thus not considered 'damaging'. It would be better to tune down this statement to something like: "We hypothesized that some of the remaining heritability could be explained by the combined burden of rare damaging gene variants".

---

## [Author Response]

Essential revisions:1) The difference between male and female longevity is one of the most conserved observations in human biology. In this light, it is important to see if any effects are sex specific or sex biases? Once the sexes are separated, is the effect observed in both or is it stronger in one sex? It would be interesting if the authors can speculate a bit more (in their Discussion section) about why they think they observed an association with mother's, but not father's age at death, given that the genetic correlation of longevity with father's age at death is much stronger than with mother's age at death (see Deelen et al., 2019). Hence, it may be that the ultra-rare PTV burden is relevant for lifespan, but not so much for longevity. A related but non overlapping question is whether PTVs have more damaging effects when on the X chromosome, where they are not compensated for by a second allele in males?

We thank reviewers for excellent questions. As suggested, we ran analysis separately for men and women and found sex-specific effects for lifespan phenotypes. Association with age at death was similar between the sexes. However, the healthspan signal was mostly driven by women. At the same time, women had a much longer healthspan compared to men (Cox PH beta = 0.16, p-value = 1.55e^-18^). We hypothesize that the genetic component, represented here by ultra-rare PTVs, may play a less important role in male healthspan due to the elevated unhealthy lifestyle choices such as smoking, drinking, risky behavior and unhealthy diet. Indeed, men are known to smoke more [1], drink more and have higher BMI scores [2] than women in UKB. Results of this analysis are summarized in Table 2.

We carefully read an excellent paper by Deelen et al. and grateful to the reviewers for bringing this discrepancy to our attention. There are various explanations for lack of association between father’s age at death and PTV burden, while genetic correlation of longevity with father’s age at death is stronger than mother’s age at death. As reviewers suggest, one of these explanations is that ultra-rare PTV burden is relevant for lifespan, but not so much for longevity. To test this hypothesis, PTV burden can be compared between centenarians and an appropriate control group. If PTV burden is associated with longevity, we would expect centenarians to be severely depleted of ultra-rare PTVs.

To address the question about X chromosome, we tested the association between ultra-rare PTV burden on X chromosome and lifespan and healthspan in men. We found no associations in both cases. However, the number of PTVs per individual specifically on X chromosome is extremely low, thus we may be out of power to pick up the difference.

We have made various changes in the manuscript on these issues.

2) Are there genes with more frequent ultra-rare PTVs than other genes based on what is explored in the paper? This could be tested with gene-based burden tests, where more frequent PTV regions should be investigated in terms of functional annotation.

Thank you for the excellent suggestion. We have tested gene burden of ultra-rare PTVs in the general population as well as in lifespan phenotypes. This analysis revealed genes with the disproportionately high PTV number in the UKB cohort to have high oe scores, andvice versa. Thus, our analysis generally recapitulates gnomAD findings. Interestingly, Fisher’s test didn’t reveal a significant increase in the burden of particular genes when compared between subjects with shorter and longer lifespan, as well as shorter and longer healthspan. These findings are now discussed in the section ‘Gene burden test’, and are supported by Figure 4—figure supplements 2-4 and Supplementary files 1-3.

3) On the contrary, the ~1500 genes without PTVs, are they also found to be "human gene knock-outs"? What are the overlap of these identified genes with those identified in other similar projects?

We addressed the overlap between 1,500 genes without PTVs in UKB and other datasets by analyzing pLI scores of these genes. pLI scores were introduced by ExAC and represent the probability of being intolerant to PTVs. Thus, the higher a pLI score is the more likely this gene is intolerant to PTVs. In agreement with the ExAC dataset, 1,500 iPTV genes in UKB had higher pLI scores than the rest of genes (Figure 4—figure supplement 1B), confirming that genes intolerant to PTV largely overlap between UKB and ExAC cohorts.

4) More info on the Cox models is needed. What do the proportional hazards look like, Schoenfeldt residuals? Describe the underlying time scale: does it matter if it is age or time since measurement? Are there time-varying effects here, for example in the somatic mutation accumulations? Provide all estimates with 95% CI’s, less focus on p-values overall in the manuscript is desired. Related, in Figure 2, why did the authors decide to use follow-up time as timescale for their Kaplan-Meier curve? It would make more sense to use the actual age of the included individuals (as done in Figure 4A) given that the age range is quite narrow.

We are grateful to the reviewers for these questions. Indeed, morbidity and mortality risk models were not discussed in sufficient detail. We considerably updated the text in the ‘Survival analysis’ of the Results. We hope that the amended version of the manuscript provides a substantially better explanation.

We confirmed limited effects of somatic mutations on mortality acceleration and described it in the ‘Somatic mutations and mortality acceleration’ section. Thus, we were able to use log-linear models of mortality risks, that are naturally proportional hazards models, including exponentials age variables. We updated the ‘Survival analysis’ section accordingly.

We applied two risk models with different underlying assumptions. The first one was a usual risk model: the estimate of the risk of death during the follow-up time (with censored events). In UKB, the age range is 51-82 with the follow-up time of 11 years. The effect size for the association of the total number of PTVs with lifespan during the follow-up was obtained from the standard Cox proportional hazards model using the explicit age, gender and 40 genetic markers as covariates.

We used the age at enrollment and the follow-up time in the same model for the following reasons. Cox proportional hazard models effectively provides the estimate of hazard function that is by design is exponential of a linear combination of covariates. In such a way, the model captures an exponential character of mortality acceleration with age (aka Gompertz law) and the corresponding proportional hazards regression coefficient is consistent with the empirical mortality doubling rate. The characteristic time scale in the model is thus nothing else but the mortality rate doubling time. The revised section of the manuscript now contains the explicit discussion of the models and the time scales.

The survival model involving the follow-up time and the explicit age as the regression parameter is a maximum likelihood estimator of probability of short-term survival for the individuals healthy enough to survive till the age of the first assessment. In this form, the survival model does not depend on the life history of the individuals prior the assessment and hence is robust with regard to enrollment bias effects. We inserted the necessary explanation into the text.

Standard Cox PH cannot be used for healthspan studies in UKB since only 28% of UKB participants are diagnosed with or experienced signs of age-related disease by the time of the first assessment. In Zenin et al., 2019, we described a maximum likelihood PH model of morbidity risk using the age of the first assessment and the age of the first diagnosis as a lifespan trait, and account for sex and genetic principal components. We employed the same model in this work. We provide explanations and references to the mathematical procedures used. We also provided CI for the PH regression variables along with p-values. This is indeed a better way to convey the results for the mortality and morbidity rate estimates.

The sex label is also included in the models as a covariate and hence the model learns the appropriate sex difference in mortality (we discuss this in the text).

As for Figure 2, the choice of the follow-up time was more natural than the age due to the character of the survival model used in the manuscript. Using the age instead of the follow-up time is possible, but may be not that useful. First, such analysis would illustrate a statistical hypothesis different from that which is behind the survival analysis. Due to Gompertz mortality acceleration, most of the death events involve the oldest individuals. Accordingly, the KM analysis here is naturally limited to a relatively narrow (much narrower than the UKB age-span) age group representing those close to the maximum age in the UKB population.

5) The section about somatic mutations and mortality acceleration is currently only based on estimations of somatic variation and computational modeling. It would be better if the authors use the actual sequencing data to try to detect somatic variants (using variant allele fractions) and subsequently associate these with lifespan and healthspan.

We thank reviewers for the excellent suggestion. Cox proportional hazards model revealed no association between somatic mutation burden and lifespan in UKB. We defined somatic mutations as variants covered by at least 80 reads with variant allele fraction between 0.05 and 0.3, thus each variant was present in a minimum of 4 reads. Variant allele fractions were calculated as ratios of AD to DP, which included in the vcf file of each UKB individual. We additionally excluded all variants with UKB frequency greater than 1% as those variants are fixed in the population and unlikely to be somatic events, and focused on the variants present within the coding regions only. Variants that passed filters were summed up for each individual and tested for the association with the follow-up lifespan using Cox proportional hazards model and corrected for age at assessment, sex, assessment center, and 40 genetic principal components reported in UKB. This analysis required accurate mitigation of different technical factors which we are unable to provide in a limited time. We observed a group of outliers with twice the number of somatic mutations compared to the rest of the population. It appears as a technical artefact, the nature of which we were unable to track down. This issue needs further investigation.

6) Are the analyses affected by the errors on mis-mapping encountered in the exome data from UKB? If so, have the authors re-ran the analyses after correcting errors?

We appreciate this question. We re-ran our analysis using corrected UKB data and updated all figures accordingly. We found that the issue has a very minor effect on our analysis.

7) Figure 4C; Did the authors only compare the 0, 1e^-4^ bin with the 1e^-4^, 1e^-3^ bin or was that the only comparison showing a significant difference? If the latter is the case, they should also report the P-values for the comparison with the other 2 bins.

We now compared every bin to the 0, 1e^-4^ bin and added p-values to the figure accordingly (Figure 4C).

8) Figure 5: The authors should add a panel with the difference in the LOF-gene between young and old (even if this is not significant), given that this one is also significantly associated with lifespan (Figure 5A).

We calculated the difference in the LOF-gene between young and old cohorts and updated Figure 5 accordingly (Figure 5E).

9) Cohort differences should be acknowledged: are these cohorts representative of the general UK population? There is some discussion around this but more is needed. What are the implications if these cohorts are prone to selection bias? Authors should discuss the possibility that data beyond the 11 years of follow-up in UKB may lead to a bias in the results even if it's not possible to account for it directly. What are the mean ages at death in UK in general, and in the two cohorts?

Thank you for the suggestion. There is indeed a possibility that a longer follow-up and older UKB cohort would show different association between lifespan and PTV burden. Having significant association in much older UKBBN cohort helps us to speculate that the association will remain significant in the UKB cohort as it ages. We updated the Discussion accordingly by adding the following text:

“It is also unclear if association won't disappear beyond 11 years of follow up in UKB. Findings from much older UKBBN cohort suggest that effect sizes of ultra-rare PTVs on lifespan will remain significant. However, to fully understand the role rare variants playing in human lifespan, we need to test its effects in older UKB cohort as well as in large ethnically diverse datasets.”

We also compared mean ages at death in the UK with UKB and UKBBN cohorts. We updated the Discussion accordingly by adding the following text:

“General UK population born at the same time (approximately a year of 1963) had a life expectancy at birth of 71 years which is much older compared to the average lifespan of 57 years in the deceased UKB cohort. On the other hand, UKBBN subjects had an average lifespan of 69 years which is much closer to the actual lifespan in the UK.”

10) The Introduction section currently contains some statements that are incorrect, and should be addressed:"For example, GWAS on centenarians consistently demonstrate the loci near LPA and APOE, FOXO3A, HLA-DQA1 and SH2B3 genes to be associated with longevity (Serbezov et al., 2018)". The majority of the loci mentioned here have consistently been associated with parental lifespan, but not with longevity (which is defined as living to an age above a certain age threshold). The only locus consistently associated with longevity (including studies containing centenarians) is APOE. Hence, the authors need to update this statement and use a more appropriate reference here, such as the recent review from Melzer et al., 2019.

Thank you very much. We have updated the Introduction according to this suggestion.

"However, the combined contribution of these variants could explain only a small part of lifespan heritability, at least as asserted from twin studies (Ruby et al., 2018)." The paper by Ruby et al. used pedigree data to estimate the heritability of lifespan. Hence, the authors cannot use this reference to refer to the heritability as estimated by twin studies and should thus update this reference or change their statement.

We appreciate the reviewers catching this inconsistency. We updated the Introduction accordingly.

"We hypothesized that the rest could be explained by the combined burden of rare damaging gene variants". Why do the authors assume that the rare variants have to be damaging? It could well be that there is a contribution of variants that lead to increased functioning of genes/proteins and are thus not considered 'damaging'. It would be better to tune down this statement to something like: "We hypothesized that some of the remaining heritability could be explained by the combined burden of rare damaging gene variants".

Thank you. We modified the text as suggested, please see the first paragraph of the Introduction.